# Integrated Transcriptome and Metabolomics to Reveal the Mechanism of Adipose Mesenchymal Stem Cells in Treating Liver Fibrosis

**DOI:** 10.3390/ijms242216086

**Published:** 2023-11-08

**Authors:** Haifeng Liu, Xinmiao Wang, Hongchuan Deng, Haocheng Huang, Yifan Liu, Zhijun Zhong, Liuhong Shen, Suizhong Cao, Xiaoping Ma, Ziyao Zhou, Dechun Chen, Guangneng Peng

**Affiliations:** 1Department of Veterinary Surgery, College of Veterinary Medicine, Sichuan Agricultural University, Chengdu 611130, China; hfliu@sicau.edu.cn (H.L.); wxm970618@163.com (X.W.); d18784998479@126.com (H.D.); hhc3267074227@163.com (H.H.); shenlh@sicau.edu.cn (L.S.); suizhongcao@126.com (S.C.); mxp886@sina.com.cn (X.M.); 13086613810@165.com (Y.L.); zhongzhijun488@126.com (Z.Z.); zzhou@sicau.edu.cn (Z.Z.); 2College of Animal and Veterinary Sciences, Southwest Minzu University, Chengdu 610041, China

**Keywords:** adipose-derived mesenchymal stem cells (ADMSCs), liver fibrosis (LF), bile acid (BA), retinol metabolism

## Abstract

Liver fibrosis (LF) is a late-stage process observed in various chronic liver diseases with bile and retinol metabolism closely associated with it. Adipose-derived mesenchymal stem cells (ADMSCs) have shown significant therapeutic potential in treating LF. In this study, the transplantation of ADMSCs was applied to a CCl_4_-induced LF model to investigate its molecular mechanism through a multi-omics joint analysis. The findings reveal that ADMSCs effectively reduced levels of alanine aminotransferase (ALT), aspartate aminotransferase (AST), total bilirubin (TBIL), gamma-glutamyltransferase (GGT), Interleukin-6 (IL-6), tumor necrosis factor-alpha (TNF-α), and α-Smooth muscle actin (α-SMA), thereby mitigating liver lesions, preventing liver parenchymal necrosis, and improving liver collagen deposition. Furthermore, 4751 differentially expressed genes (DEGs) and 270 differentially expressed metabolites (DMs) were detected via transcriptome and metabolomics analysis. Conjoint analysis showed that ADMSCs up-regulated the expression of *Cyp7a1*, *Baat*, *Cyp27a1*, *Adh7*, *Slco1a4*, *Aldh1a1,* and *Adh7* genes to promote primary bile acids (TCDCA: Taurochenodeoxycholic acid; GCDCA: Glycochenodeoxycholic acid; GCA: glycocholic acid, TCA: Taurocholic acid) synthesis, secretion and retinol metabolism. This suggests that ADMSCs play a therapeutic role in maintaining bile acid (BA) homeostasis and correcting disturbances in retinol metabolism.

## 1. Introduction

Any cause of chronic liver disease (hepatitis B and C, alcohol, cholestasis, and non-alcoholic steatohepatitis (NASH)) leads to LF [1]. LF is characterized by the excessive synthesis and deposition of the extracellular matrix (ECM) [2]. Hepatic stellate cells (HSC) are considered major effectors of LF because they are involved in ECM deposition and trigger immune responses by secreting cytokines and chemokines. In the process of HSC activation, on the one hand, HSC can stimulate the secretion of IL-17 in TH17 cells, up-regulate the expression of NF-κB and STAT3, and aggravate liver inflammation [3]. On the other hand, the inhibition of the CXCR7-Id1 pathway through the fibroblast growth factor receptor 1 (FGFR 1)-CXCR4 axis inhibits normal liver regeneration and aggravates LF [4]. The activation of HSC is accompanied by a large loss of retinoids, which further aggravates inflammation and fibrosis progression and accelerates the occurrence of liver tumors. Unfortunately, the biological function of retinoid loss in HSC and LF remains unclear. Meanwhile, in liver injury, *Cyp8a1* downregulation inhibits the intrahepatic synthesis of BA and exacerbates LF [5]. The inactivation of the farnesoid X receptor (FXR) after liver injury leads to an increased BA synthesis rate and fibroblast growth factor 19 (FGF19) expression in the liver, which induces the activation of inflammatory pathways and promotes hepatocyte apoptosis, leading to LF [6,7]. Therefore, inhibiting HSC activation, regulating the level of body and immune response, and correcting metabolic disorders are effective ways to prevent or reduce LF. Unfortunately, there is no generally accepted therapy for the treatment of LF.

ADMSCs have attracted great interest in the field of disease treatment due to their multi-directional differentiation potential and self-renewal ability [8]. The transplantation of ADMSCs has been shown to be effective in the treatment of colitis by inducing a macrophage phenotype shift from M1 to M2 [9]. In addition, extracellular vesicles released by ADMSCs attenuate neuroinflammation associated with Alzheimer’s disease via miR-122 [10]. Simultaneously, numerous studies have demonstrated that the transplantation of ADMSCs exhibits therapeutic efficacy in diverse systemic ailments, including kidney, heart, and bone diseases [11,12]. The liver, which is the largest endocrine organ, is susceptible to damage caused by various pathogenic factors, resulting in metabolic disturbances. Studies have shown that transplanted ADMSCs are attracted to damaged liver tissue by various inflammatory factors and chemokines. They promote endogenous hepatocyte regeneration and maintain the normal structure of the liver by secreting cytokines such as the hepatocyte growth factor (HGF) and FGF [13,14]. In the context of liver ischemia-reperfusion injury, ADMSCs have been found to play a role in inhibiting hepatocyte apoptosis. This inhibition is achieved through the up-regulation of the expression of Fas, caspase-3, and caspase-9. Additionally, ADMSCs also demonstrate the ability to suppress hepatocyte autophagy via upregulating superoxide dismutase (SOD) activity [15,16]. Studies have shown that the transplantation of ADMSCs can effectively treat LF by promoting liver differentiation, inhibiting inflammatory response, and suppressing HSC activation [17]. These effects are primarily achieved through the inhibition of LF factors such as nerve growth factor (NGF), transforming growth factor-β1 (TGF-β1), α-SMA, hyaluronic acid, and hydroxyproline [17,18,19]. In clinical practice, the transplantation of ADMSCs has been reported to treat patients with liver cirrhosis [20]. Most studies have demonstrated that the transplantation of ADMSCs can effectively treat liver diseases by regulating the immune system, suppressing inflammation, and reducing LF markers. However, there is currently no report on whether the transplantation of ADMSCs can treat LF by modulating liver metabolic pathways.

Metabolomics is a burgeoning research method that focuses on analyzing endogenous metabolites found in the urine, serum, and tissues [21]. Its aim is to gain a comprehensive understanding of the metabolic pathways within an entire organ or system. Metabolites, which are the products of cellular regulatory processes, serve as valuable indicators of disorders resulting from drugs or diseases. By identifying the metabolic pathways in which differential metabolites are located, metabolomics provides insight into the disrupted metabolic processes associated with disease [22]. The liver, as the largest organ in the body, performs numerous synthetic and secretory functions, making it essential for the body’s metabolism [8]. Metabolomics has gained significant popularity in the study of liver diseases due to its effective qualitative and quantitative analysis of metabolites [22,23]. Transcriptomics involves the comprehensive collection of transcriptional information from tissues or organs, enabling the prediction of the potential mechanisms of external substances that stimulate metabolic pathways [24]. Numerous studies have utilized transcriptomics to analyze the gene expression profiles of different liver diseases and predict potential therapeutic targets or toxic effects of drugs on the liver [25,26]. By integrating transcriptomics and metabolomic profiles, researchers can gain a more reliable understanding of metabolic alterations and further investigate the underlying biological processes and toxic mechanisms. Therefore, this experiment aims to explore the mechanism by which ADMSCs affect LF at the transcriptional and metabolic levels.

## 2. Results

### 2.1. ADMSCs Alleviate Liver Injury and Fibrosis in Rats

To investigate the effect of ADMSCs on LF models, a rat LF model was established via the intraperitoneal injection of CCl_4_, and ADMSCs were used for treatment. Serum biochemistry and Elisa results showed that the expression levels of liver function indicators ALT, AST, TBIL, GGT and inflammatory factors, such as IL-6 and TNF-α, were significantly increased in rats with LF (*p* < 0.05, Figure 1A–F). However, the expression of the above indicators was significantly downregulated after the transplantation of ADMSCs (*p* < 0.05). HE and MASSON staining revealed that fibrosis rats exhibited significant liver cell structural damage, intercellular lipid droplets and vacuoles, and collagen fiber deposition compared to the Cont group.

However, a significant improvement occurred after ADMSCs treatment (Figure 1I,J). Immunohistochemical methods were used to detect the expression of the LF marker α-SMA. The expression of α-SMA in the liver tissue of the CCl_4_ group was significantly increased compared to the Cont group. The transplantation of ADMSCs eliminated these protein expression imbalances (Figure 1G,H).

### 2.2. Changes in Liver Metabolic Profile after Adipose Mesenchymal Stem Cell Transplantation

Three groups of rat liver metabolites were detected using UHPLC-MS/MS, and a total of 1020 metabolites were annotated through a reliable dataBAe. The results of the OPLS-DA analysis indicated significant differences in liver metabolic product patterns between the CCl_4_ group and the other two groups, while there were some similarities in liver metabolic products between the Cont and MSC groups (Figure 2A,B).

In all three experimental groups, six biological repeats were conducted, resulting in the identification of 267 DMs (Figure 2C). These findings suggest that there are significant differences in metabolic status among the groups. The present study identified significant differential metabolites in the liver tissue of Cont, CCl_4_, and MSC rats using pairwise comparisons. It was found that, compared to the Cont group, the CCl_4_ and MSC groups had 146 (65 up- and 81 downregulated) and 142 (86 up- and 56 downregulated) DMs in the liver tissue, respectively. Additionally, the CCl_4_ and MSC groups had a total of 161 (96 up- and 65 downregulated) DMs in the liver tissue (Figure 2D). This study found that there are 84 types of DMs expressed in both Cont vs. CCl_4_ as well as MSC vs. CCl_4_ with opposite trends (Appendix A). These findings indicate that the treatment of ADMSCs may have the ability to reverse some of the biological functions that are damaged by CCl_4_.

In order to reveal the specific mechanism of ADMSCs participating in the treatment of LF, the KEGG dataBAe was used to conduct pathway enrichment analysis on DMs in three groups. The top 10 metabolites up- and downregulated in CCl_4_ vs. Cont and MSC vs. CCl_4_ are shown in Figure 2F,G. Compared with the Cont group, 4 of the top 10 up-regulated metabolites belong to amino acids (Thiazolidine-4-carboxylic acid, Arginine, Stachydrine and G-guanidinobutyrate) and 4 of the top 10 downregulated metabolites belong to BA (Lithocholylglycine, Glycocholate, Glycochenodeoxycholate and Glycodeoxycholic acid) (Figure 2F) in CCl_4_ group. As two markers of bile secretion, the expression of Glycocholate and Glycochenodeoxycholate in CCl_4_ vs. Cont decreased significantly. After MSC treatment, this trend was reversed (Figure 2F,G). According to the *p*-value of KEGG enrichment (Figure 2E,H), it was found that most DMs are closely related to the metabolism and secretion of the bile pathway, including mainly retinol metabolism, primary bile biosynthesis, cholesterol metabolism, and beta-Alanine metabolism, etc.

### 2.3. Changes in Liver Transcriptome after Adipose Mesenchymal Stem Cell Transplantation

RNA sequencing (Rna-Seq) technology was used to detect the gene expression in liver tissues of the Cont group, CCl_4_ group, and MSC group. Sequencing data were quality-controlled and annotated, and all sequencing results matched more than 95% of the reference genome (Appendix A). Principal component analysis (PCA) showed that the gene pattern of rats in the CCl_4_ group was clearly separated from the remaining two groups. In addition, we found similar gene expression patterns in the liver of rats in the Cont and MSC group (Figure. 3A). A total of 4751 DEGs were identified through pairwise comparison (Figure 3B and Appendix A). Compared with Cont rats, 3669 genes (2286 upregulated and 1383 downregulated) and 1112 genes (925 upregulated and 187 downregulated) were differentially expressed in CCl_4_ and MSC rats, respectively. In addition, 2747 DEGs (1185 upregulated and 1562 downregulated) were observed between CCl_4_ and MSC rats (Figure 3C–E).

The pathway enrichment analysis of detected DEGs was conducted BAed on the KEGG dataBAe. According to the *p*-value of KEGG enrichment, the metabolism of xenobiotics via cytochrome P_450_, drug metabolism cytochrome P_450_, steroid hormone biosynthesis, fatty acid degradation, the PPAR signaling pathway, Retinol metabolism, tryptophan metabolism, and butanoate metabolism are the common pathways of CCl_4_ and ADMSCs. Primary BA biosynthesis and bile secretion were enriched, which further proves that the liver function of ADMSCs is related to bile homeostasis (Figure 3F,G).

### 2.4. Alterations of Genes and Metabolites Related to Primary BA Biosynthesis and Bile Secretion

In order to further elucidate the relationship between the two omics results, conjoint analysis was conducted BAed on the KEGG enrichment results, with the most important being the synthesis of primary BA and bile secretion. As shown in Figure 4A,C, compared to Cont and CCl_4_, the DMs and DEGs present in these two pathways in MSC exhibited similar and almost opposite expression trends, indicating that ADMSCs may treat LF by regulating the primary BA biosynthesis and bile secretion disturbances caused by CCl_4_.

An important BA, Glycochenodeoxycholic acid (GCDCA), was downregulated in rats with LF because the genes *Slc27a5*, *Amacr*, *Acox2*, *Scp2*, *Baat*, *Acnat1*, and *Acnat2* involved in primary BA synthesis were all downregulated (Figure 4A,B). The expression of *Cyp7a1* and *Akr1d1* was not significantly downregulated in rats with hepatic fibrosis, but this trend was significantly reversed after the transplantation of ADMSCs (Figure 4B). *Cyp27a1* and *Hsd17b4* have opposite trends to the two genes described above. After ADMSCs treatment, not only did the expression of GCDCA and GCA increase, but two important BA, Taurochenodeoxycholic acid (TCDCA) and Taurocholic acid (TCA), were also detected but were not found in CCl_4_ vs. Cont (Figure 4B). When the bile secretion pathway was activated after primary BA synthesis, the four representative primary BA were transported to hepatocytes. At the same time, the expression levels of genes *Slco1b2*, *Slco1a4*, *Slc10a1*, and *Ephx1* in MSC vs. CCl_4_ were increased (Figure 4D). BA maintain cholesterol balance. The increased expression of *Cyp7a1* after the transplantation of ADMSCs accelerated the conversion of cholesterol to BA, and the upregulation of *Nr1h4* promoted the synthesis, transport, and detoxification of BA (Figure 4C). The genes *Nr1h4*, *Slc27a5*, *Ugt1a6*, *Ugt2b35*, and *Baat*, which alleviate BA toxicity, were upregulated in the MSC group (Figure 4D). Meanwhile, the increased expression of *Abcg8*, *Abcg5*, and *Abcc2* in the MSC group promoted the excretion of BA into bile ducts (Figure 4D).

### 2.5. Alterations of Genes and Metabolites Related to Retinol Metabolism

In addition to primary BA synthesis and bile secretion, retinol metabolism also plays an important role in the regulation of metabolic disorders caused by CCl_4_-induced LF [27]. There are three important retinol metabolites, Retinal, 4-Oxoretinol, and 9-cis-Retinal, which are downregulated in CCl_4_ vs. Cont, while in MSC vs. CCl_4_, their expression is upregulated, suggesting that ADMSCs may treat CCl_4_-induced LF by promoting retinol metabolism (Figure 4E,F). Genes associated with retinol metabolism showed similar expression patterns in Cont and MSC. Figure 4E shows that, among the genes most related to retinol metabolism, there were 17 genes belonging to the cytochrome P_450_ family-1,2,4,3,26, all of which played important roles in retinol metabolism.

### 2.6. Correlations between DGEs and DMs Related to Primary BA Biosynthesis, Bile Secretion and Retinol Metabolism

BAed on these aforementioned findings, it can be observed that the regulation of a metabolite can be influenced by a diverse array of genes, subsequently impacting and controlling various pathways. Consequently, the association between DEGs and DMs holds significant importance for the identification of subsequent biomarkers. To this end, we involved a correlation network of primary BA synthesis, bile secretion, and retinol metabolism pathways in MSC vs. CCl_4_. Notably, all DEGs and DMs included in the network diagrams exhibited correlations exceeding 0.7 and possessed statistical significance.

A total of 35 genes and 8 metabolites were identified that jointly regulated the primary BA biosynthesis and BA secretion pathways (Figure 5A). Among these genes, *Acox2* was specifically related to TCDCA. Furthermore, GCDCA is associated with *Slc27a5*, *Baat*, and *Slco1a4*, while GCA is linked to *Cyp7a1*, *Abcb4*, *Akr1d1*, and others. These associations suggest that multiple genes may co-regulate the aforementioned metabolites. In addition, retinol metabolism, as a key pathway in LF, is regulated at the transcriptional level via ADMSCs, where *Ugt2b35*, *Cyp3a18*, *Cyp2c13*, *Retsat*, *Adh7*, and *Dhrs4* are related to the metabolic markers 4-Oxoretinol, Retinal, and 9-cis-Retinal (Figure 5B).

### 2.7. Candidate Gene Validation

Subsequently, the expression levels of seven key genes involved in BA biosynthesis, bile secretion, and retinol metabolism were verified through qPCR and Western blotting (Figure 6). The primers for these seven genes are listed in Table 1. Notably, the expression levels of these genes decreased after CCl_4_ intervention but were significantly upregulated following the treatment of ADMSCs. It is interesting that the expression levels of these genes in the liver tissue of rats treated with ADMSCs were higher than those in the Cont group, and there was a statistical difference compared to the expression levels of related genes in CCl_4_. Overall, these results are consistent with transcriptome sequencing findings, suggesting that ADMSCs play a therapeutic role in regulating the expression of key genes in relevant pathways.

## 3. Discussion

LF, a wound-healing response, is a common occurrence in various liver injuries and poses a significant threat to both human and animal health. One highly toxic compound, CCl_4_, is known to cause liver necrosis and release proinflammatory and fibrotic cytokines, thereby leading to LF and potentially cirrhosis [28]. Several studies have indicated that a disruption in the metabolism of BA (BA) can contribute to liver injury and fibrosis. Various interventions, such as inhibiting toxic BA levels and regulating BA homeostasis using Taurocholic acid and the probiotic Lactobacillus rhamnosus GG, have shown promising results in improving liver injury and fibrosis in mice [29]. While ADMSCs have shown therapeutic effects on CCl_4_-induced liver injury models [30,31], it is still unclear how they exert these effects by regulating metabolic levels in the body. This study aims to identify and analyze the metabolic and transcriptional products of liver tissues in rats with CCl_4_-induced LF and investigate the specific mechanism through which ADMSCs treat LF.

BA (BA) are amphiphilic molecules that are produced through the metabolism of cholesterol in the liver. They make up over 50% of the organic matter in bile, and their synthesis and secretion pathways are intricately regulated. BA consist of the following two components: primary BA, which are produced directly by hepatocytes, and secondary BA, which are formed after being secreted into the intestine [32,33]. Secondary BA synthesized in the intestine are absorbed via the liver through the hepatic portal vein, playing a crucial role in maintaining the balance of bile flow in the body. In the liver, these BA contribute to the clearance of endogenous compounds and metabolites, such as bilirubin and hormones, as well as exogenous substances like drugs. Moreover, BA serve as signaling molecules that regulate lipid, glucose, and energy metabolism in the liver. They achieve this by interacting with the BA receptor, FXR, and takeda G protein-coupled receptor 5 (TGR5). Furthermore, secondary BA in the intestine play a significant role in controlling cell proliferation and inflammatory responses in both the liver and intestine. This is achieved through the absorption of lipophilic nutrients into the intestine [33]. The occurrence of a variety of liver diseases is accompanied by the disorder of BA metabolism [34,35]. In the CCl_4_-induced liver injury model, the synthesis, secretion, and reabsorption of BA are disrupted, leading to the increased expression of total bile acid (TB) in the liver and promoting the progression of LF [36,37]. Studies have found that the accumulation of BA in hepatocytes accelerates the progression of NASH by promoting ER stress and de novo fat synthesis. In HFD-induced NAFLD, researchers found that the expression of Ursodeoxycholic acid (UDCA) and Taurohyodeoxycholic acid (THDCA) in the liver was significantly decreased [38]. In addition, the gut microbiota-mediated inactivation of the FXR-FGF15 axis was observed in patients with primary-sclerosed cholangitis (PSC) due to cholestasis, resulting in the persistently elevated content of BA in the liver through the negative feedback regulation of BA’ synthesis inhibition [39]. In dimethylnitrosamine (DMN)-induced LF, the expression of TCDCA, GCDCA, GCA, and TCA in the liver was significantly reduced [40]. TCDCA has the ability to resist cell apoptosis and reduce endoplasmic reticulum stress and has demonstrated its therapeutic potential in experimental acute liver injury induced by excessive acetaminophen (APAP) [41]. When the content of GCDCA is decreased, the presence of cholestatic liver disease is often suggested [42]. In this study, the four BA detected were TCDCA, GCDCA, TCA, and GCA, all of which belong to primary BA. This indicates that ADMSCs mainly exert therapeutic effects by affecting the synthesis of primary BA.

BA are primarily produced in the liver through two enzymatic pathways as follows: the classical or neutral synthesis pathway, which involves the rate-limiting enzyme *Cyp7a1* [43], and the acidic or remedial pathway, initiated by *Cyp27a1* [44]. The inhibition of either pathway results in impaired bile flow in the body. It is important to note that in rats with LF, the expression of *Cyp27a1* is significantly reduced. However, after treatment with ADMSCs, the expression of *Cyp7a1* is significantly increased. This suggests that CCl_4_ may hinder the synthesis and secretion of BA by inhibiting the alternative pathway of bile synthesis, while ADMSCs promote stable bile flow by enhancing the classical synthesis pathway. *Slco1a4*, as a member of the OATP family, is involved in the liver’s absorption of free secondary BA from the blood [45]. The Solute Carrier Family 27 Member 5 gene (*Slc27a5*) is a member of the *Slc27a* gene family and encodes the fatty acid (FA) transport protein 5 (*Fatp5*). This protein plays a crucial role in FA transport and BA metabolism. Studies have shown that *Slc27a5* knockout mice have reduced FA intake, accompanied by liver injury, insulin resistance, and dyslipidemia [46,47]. The primary free BA (CDCA, CA) produced by the liver needs to be conjugated with glycine or taurine to become primary conjugated BA (TCDCA/GCDCA/TCA/GCA), which enters the intestine. This process is regulated by *Baat* and is often accompanied by a decrease in *Baat* expression in cholestatic liver diseases [48,49]. Previous research indicates that MSCs upregulate the expression of *Slc27a5* and *Baat*, promoting the synthesis and secretion of BA, as well as lipid reabsorption. Additionally, *Abcg5* and *Abcg8*, which are part of the ATP-binding box (ABC) transporter family, play a crucial role in promoting cholesterol excretion into bile in hepatocytes [50]. Another metabolite involved in these pathways is Glutathione (GSH), which has been reported to regulate sterol balance by affecting the activity of Cholesterol 7 alpha-hydroxylase (CH-7 alpha) [51].

The activation of hepatic stellate cells plays a pivotal role in the progression of LF, as their transition into myofibroblast-like cells serves as the primary mechanism for the deposition of the ECM. These cells also serve as an important reservoir for vitamin A, and their activation can result in the significant depletion of this essential nutrient. Vitamin A encompasses biologically active compounds such as retinol and its metabolites, including Retinal, 9-cis-retinal (9cRAL), and 4-oxidized retinol, which were identified in this study. Vitamin A plays an important role in the body’s vision, growth [52], reproduction, immune function [53], and metabolic processes. The genes *Adh7*, *Rdh5*, *Retsat*, and *Aldh1a1*, which play a crucial role in the metabolic pathway of vitamin A [54], specifically, *Adh7* and *Aldh1a1*, facilitate the transformation of retinol into its biologically active form, retinoic acid (RA) [55]. Many studies have shown that liver cells in vitamin A-deficient rats show obvious vacuolization, steatosis, and mild inflammatory infiltration [56]. In addition, it was found that the progression of LF was negatively correlated with liver RA content. Vitamin A enters the liver only through the transport of secondary BA in the intestine. Studies have shown a correlation between VAD, cholestasis, and fibrosis in patients. This association can be attributed to two primary factors. First, reduced bile flow from the liver leads to lower concentrations of BA in the gut, which hinders the absorption of vitamin A. Second, alterations in the gut microbiota prevent the hepatic uptake of secondary BA by inhibiting FXR signaling. In our study, we found that the synthesis of four primary BA (TCDCA, GCDCA, TCA, and GCA) was inhibited in CCl_4_-induced fibrosis rats, while the levels of retinol-related secondary metabolites were significantly reduced. However, the demyelination trend was reversed after the transplantation of ADMSCs. These findings indicate that ADMSCs may possess therapeutic properties by suppressing HSC activation and enhancing bile flow, ultimately facilitating the absorption of vitamin A.

BAed on the results of this experiment, we found that ADMSCs promote the synthesis and secretion of TCA, GCA, TCDCA, GCDCA, and GSH. Additionally, they enhance the expression of key genes (*Cyp7a1*, *Cyp27a1*, *Abcg5*, *Abcg8*, *Baat*, and *Slc27a5*) associated with these metabolites, thereby promoting sterol balance and bile secretion in the body. On the other hand, ADMSCs upregulate the expression of key genes such as *Adh7*, *Slco1a4*, *Aldh1a1*, *Retsat*, and *Dhrs4*, which helps maintain the stability of Retinal, 9-cis retinol, and 4-Oxoretinol in the liver. This inhibits the activation of hepatic stellate cells and promotes an immune response, leading to therapeutic effects.

## 4. Materials and Methods

### 4.1. Animal Treatment and Sample Collection

Eighteen male Sprague Dawley rats, 4 weeks old, weighing 200–220 g, were purchased from the Dashuo Experimental Animal Company (Chengdu, China). After one week of adaptive feeding, we randomly divided them into the following three groups: Cont (control group), CCl_4_ (CCl_4_ intervention group), and MSC (ADMSCs treatment group), with six rats in each group. This study complied with the guidelines for the Care and Use of Laboratory Animals by the National Institutes of Health. This study was approved by the Sichuan Agricultural University Institutional Animal Care and Use Committee (No. SYXK 2019-187).

The control group, the CCl_4_ intervention group and ADMSCs treatment group were treated as follows: control group, intraperitoneal injection of olive oil 2 mL/kg twice a week for 4 weeks; CCl_4_ intervention group, intraperitoneal injection of CCl_4_ (2 mL/kg) mixed with olive oil solution (*v*:*v*/4:6) twice a week for 4 weeks; ADMSCs treatment group, 24 h after the last intraperitoneal injection of CCl_4_-olive oil solution, ADMSCs (1.5 × 10^6^ ADMSCs dissolved in 0.5 mL PBS solution) were transplanted via the tail vein once a week for 1 week. At the end of the animal experiment, and after fasting for 24 h, rats were anesthetized with sodium pentobarbital, and the liver and venous serum samples were collected. Serum samples were used to determine biochemical markers of liver function, and tissue samples were frozen in liquid nitrogen and stored at −80 °C for subsequent experiments.

### 4.2. Serum Biochemical and Inflammatory ELISA Factor Analysis

After blood collection from the abdominal aorta, it was immediately placed in a 4 °C ice box for one hour. Subsequently, the blood sample was centrifuged at 4 °C and 5000 r/mp for 20 min to obtain the serum. The levels of ALT, AST, TBIL, and GGT were then measured using the IDXX Biotechnology Co., Ltd. (Westbrook, Maine, USA) kit. The expression levels of inflammatory factors IL-6 and TNF-α were measured according to the manufacturer’s instructions (Ke Xing, Shanghai, China).

### 4.3. Histological and Immunohistochemistry Analysis

Liver tissue was fixed by 4% Paraformaldehyde at 4 °C for 24 h, then paraffin-embedded liver tissue was dehydrated and cut into 5 μm sections for HE and MASSON staining. The immunohistochemical determination of α-SMA expression in liver tissue was performed on the remaining embedded tissues. ImageJ software (V1.8.0.112) was used for the quantitative analysis of collagen fibers and α-SMA in MASSON and immunohistochemistry.

### 4.4. Transcriptomic Analysis

The total RNA was extracted from the liver using TRIzol (Magen, Shanghai, China). RNA samples were detected BAed on the A260/A280 absorbance ratio with a Nanodrop ND-2000 system (Thermo Scientific, Rockford, IL, USA), and the RIN of RNA was determined using an Agilent Bioanalyzer 4150 system (Agilent Technologies, Santa Clara, CA, USA). Only qualified samples were used for library construction. Paired-end libraries were prepared using a ABclonal mRNA-seq Lib Prep Kit (ABclonal, Wuhan, China). Library quality was assessed on an Agilent Bioanalyzer 4150 system, and the library preparations were sequenced on an Illumina Novaseq 6000. The original data were filtered via Q30, and low-quality fragments were eliminated using fastp software (v0.17.0). (The clean reading was then compared with the rat reference genome using HISAT2 (v2.1.0) (http://daehwankimlab.github.io/hisat2/ (accessed on 15 June 2023)) software, and all subsequent analyses were conducted on the clean reading; the FPKM algorithm was used to quantify gene expression levels. Genes with |log_2_FC| > 1 and *P*adj < 0.05 were considered DEGs. DEGs were then used for gene ontology (GO) function analysis and the Kyoto Encyclopedia of Genes and Genomes (KEGG) pathway analysis.

### 4.5. Metabolomic Analysis

We then added 200 μL of ddH_2_O to 80 mg of liver tissue for the preparation of the tissue homogenate while adding 800 μL of methanol/acetonitrile (1:1, *v*/*v*) to the homogenate. This was then centrifuged for 20 min (14,000× *g*, 4 °C) before the supernatant was dried in a vacuum centrifuge. The obtained dried substance was remelted in 10 μL of solvent acetonitrile/water (1:1, *v*/*v*), centrifuged for 15 min (14,000× *g*, 4 °C), and the supernatant was taken and loaded into a dedicated vial for UHPLC-MS/MS analysis. The samples were separated on an Agilent 1290 Infinity LC system (UHPLC)HILIC column. The specific parameters were as follows: was column temperature was 25 °C; the flow rate was 0.5 mL/min; and the injection volume was 2 μL. Mobile phase composition A was as follows: water +25 mM ammonium acetate +25 mM ammonia water, B: acetonitrile. The gradient elution procedure was as follows: 0–0.5 min, 95% B. From 0.5 to 7 min, B changed linearly from 95% to 65%. From 7 to 8 min, B changed linearly from 65 to 40%. At 8–9 min, B was maintained at 40%. From 9 to 9.1 min, B changed linearly from 40 to 95%. At 9.1–12 min, B was maintained at 95%. The samples were placed in an autosampler at 4 °C throughout the analysis. QC samples were randomly inserted into the assay to monitor the reliability of the system and experimental data. Subsequent mass spectrometry was performed on a Triple TOF 6600 mass spectrometer using electrospray ionization (ESI) in positive ion and negative ion modes. Ion source, turbo spray; source temperature, 500 °C; ion spray voltage floating (ISVF) ± 5500 V; Gas1, Gas2, and CUR were set at 60 psi, 60 psi, and 30 psi, respectively. The primary mass-to-charge ratio detection range was 60 to 1000 Da, the scan cumulative time was 0.20 s/spectra, the detection range of the second-order daughter ion mass-to-charge ratio was 25–1000 Da, and the scanning accumulation time was 0.05 s/spectra.

### 4.6. Data Processing and Annotation

Progenesis XCMS software (v 4.0.0) was used for LC-MS data extraction, alignment, peak picking, and retention time adjustment. The company’s own library with the standard mass spectra on the commercial dataBAes was compared using mzCloud (https://www.mzcloud.org (accessed on 20 May 2023)), HMDB (http://www.hmdb.ca (accessed on 20 May 2023)), and KEGG (http://kegg.jp (accessed on 22 May 2023)), All metabolites were identified by “score”, “fragment score”, “mass error (ppm)” and “isotopic similarity” [57].

### 4.7. Statistical Analysis

Data were represented as the mean ± standard deviation. The double-tailed student *t*-test, one-way analysis of variance (ANOVA), and Bonferroni post-tests were performed. PCA and orthogonal partial least squares discriminant analysis (OPLS-DA) were used to analyze metabolomic data. Permutation tests were conducted to ensure the stability of the model. The importance of each variable in the projection (VIP) value of the OPLS-DA model was calculated to indicate its contribution to classification. Metabolites with VIP > 1 and *p* < 0.05 were considered as significantly different metabolites (DMs). The Pearson correlation coefficient was used to explain the correlation between these two variables. Differential metabolites were then summarized and mapped to biochemical pathways using KEGG enrichment analysis. The pathways of significant enrichment were identified using Fisher’s precise test.

### 4.8. Validation of DEGs by qPCR and Western Blot

The total RNA was extracted from the liver using an animal total RNA isolation kit (Foregene, Chengdu, China). The cDNA was reverse transcribed using the FastKing-RT SuperMix kit (Tiangen, Beijing, China) according to the manufacturer’s protocol. In addition, qPCR was performed using the BioRad Laboratories CFX Connect™ real-time PCR Detection System. The housekeeping gene GADPH was used for the normalization of the data. The results were quantified using the 2^−ΔΔCt^ method relative to the housekeeping gene actin. The sequences of the primers used are placed in Table 1. 

Liver tissue homogenates were sonicated to dissolve completely and then centrifuged at 12,000 rpm for 30 min at 4 °C to separate the membrane-containing fraction (pellet) from the cytosol. Proteins (100 μg) were separated by 10% SDS-polyacrylamide gel electrophoresis. The separated proteins were transferred onto a nitrocellulose membrane (NC membrane, Millipore, Billerica, MA, USA), washed for 10 min with TBST, and then immersed in a blocking buffer containing 5% nonfat dry milk in TBST for 1 h at room temperature. The membrane was washed with TBST and incubated overnight at 4 °C with polyclonal primary antibodies Dhrs4 (Abcam, Cambridge, UK) diluted 1:2000, Baat (Abcam) and Cyp7a1 (Proteintech Group, Inc., Chicago, IL, USA) diluted 1:500, and Adh7 (Shanghai-Youke Biotechnology, Inc., Shanghai, China) diluted at 1:1000 in 5% nonfat dry milk. Then, the membrane was incubated with a secondary antibody (Cell Signaling Technology, Danvers, MA, USA) for 1h at room temperature. Finally, the membrane was scanned using the Odyssey Infrared Imager (LI-COR, Lincoln, NB, USA), and the bands were quantified by densitometry using Odyssey software (version 1.2, LI-COR).

## 5. Conclusions

The research results show that ADMSCs have therapeutic effects on LF induced by CCl_4_. The joint analysis of transcriptome and metabolomics revealed that ADMSCs accelerated the synthesis and secretion of primary BA (TCDCA, GCDCA, GCA, and TCA) by promoting gene expression such as *Cyp7a1*, *Baat*, *Cyp27a1*, *Adh7*, and *Slco1a4,* maintaining the body’s BA homeostasis and correcting vitamin A metabolism disorders, thus exerting therapeutic effects. The discovery of these new biomarkers in metabolomics and transcriptomics could provide new drug discovery and therapeutic strategies for LF.

## Figures and Tables

**Figure 1 ijms-24-16086-f001:**
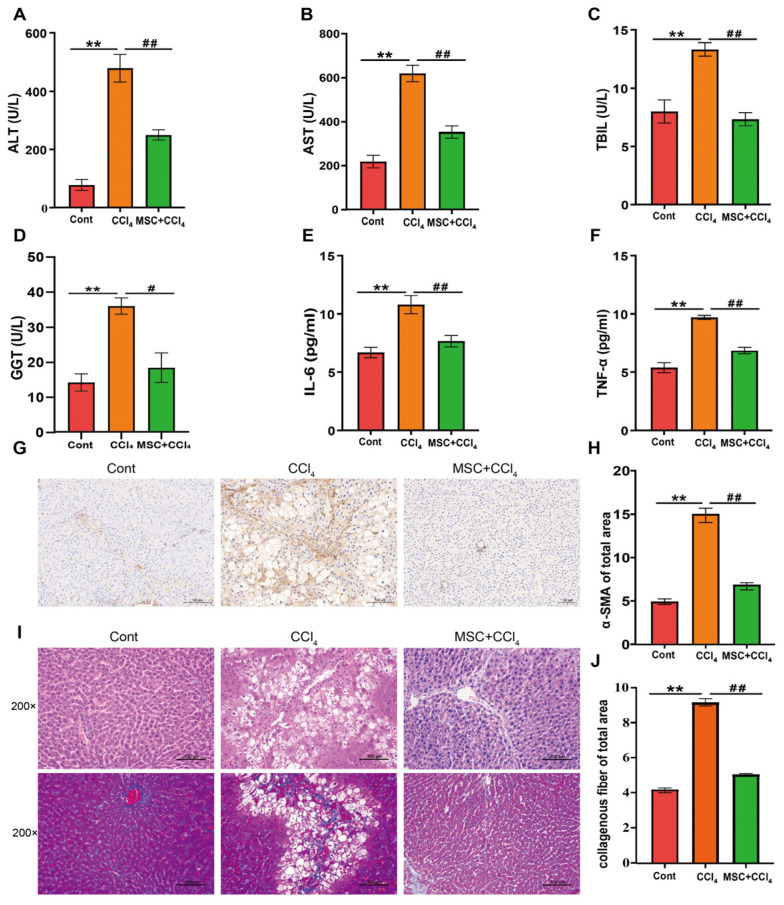
ADMSCs alleviate liver injury and fibrosis in rats. (**A**–**D**). Biochemical indicator tests of serum (ALT, AST, TBIL, and GGT); (**E**,**F**). Detection of serum Elisa-related inflammatory indicators (IL-6 and TNF-α); (**G**,**H**). Immunohistochemistry and the surface ratio of LF marker factor α-SMA (×200 magnification). (**I**,**J**). H&E staining (upper) (×200 magnification) and MASSOM staining (bottom) (×200 magnification) of the liver in different groups. Each experiment was carried out at least three times, and the data were displayed using the mean ± SD. Cont, control group; CCl_4_, CCl_4_, CCl_4_ intervention group; MSC, ADMSCs treatment group. **, *p* < 0.01 vs. Cont group; #, *p* < 0.05; ##, *p* < 0.01 vs. MSC group.

**Figure 2 ijms-24-16086-f002:**
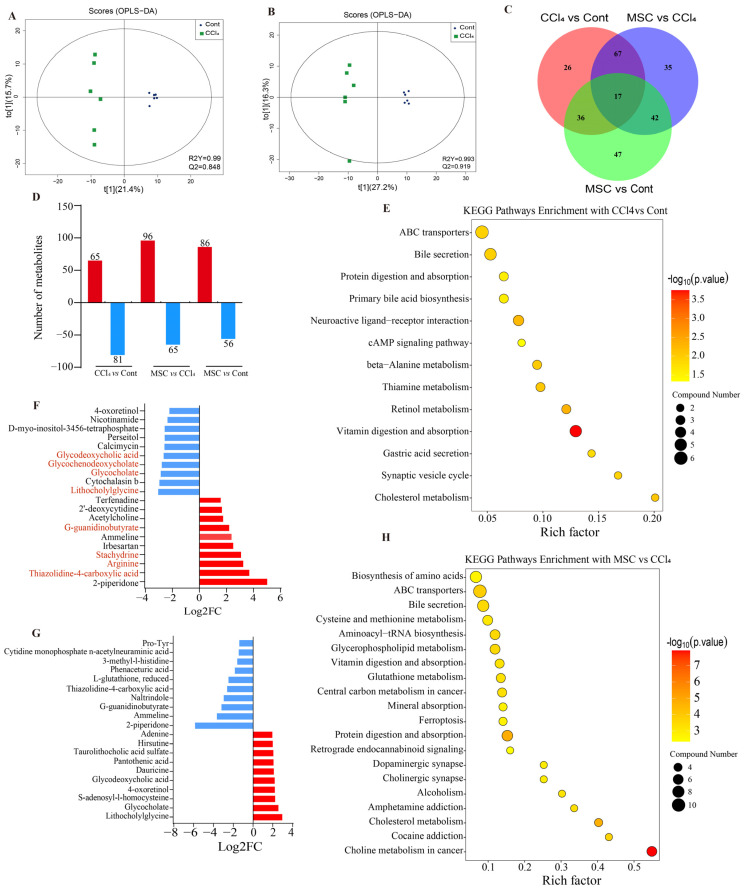
Changes in the liver metabolic profile in CCl_4_-induced LF rats and ADMSCs transplantation rats. (**A**) OPLS-DA score plot of liver metabolites after intraperitoneal CCl_4_ injection; (**B**) OPLS-DA score plot of liver metabolites after ADMSCs transplantation; (**C**) VEEN plot of the number of DMs in rat livers obtained via a pairwise comparison between Cont, CCl_4_, and MSC groups; (**D**) Number of DMs up- and downregulated in liver tissues revealed via pairwise comparisons; (**E**,**F**) Top 10 up- and downregulated DMs and KEGG enrichment of CCl_4_ vs. Cont; (**G**,**H**) Top 10 up- and downregulated DMs and KEGG enrichment in MSC vs. CCl_4_; Cont, control group; CCl_4_, CCl_4_-treated group; MSC, ADMSCs-treated group; DMs, differently expressed metabolites.

**Figure 3 ijms-24-16086-f003:**
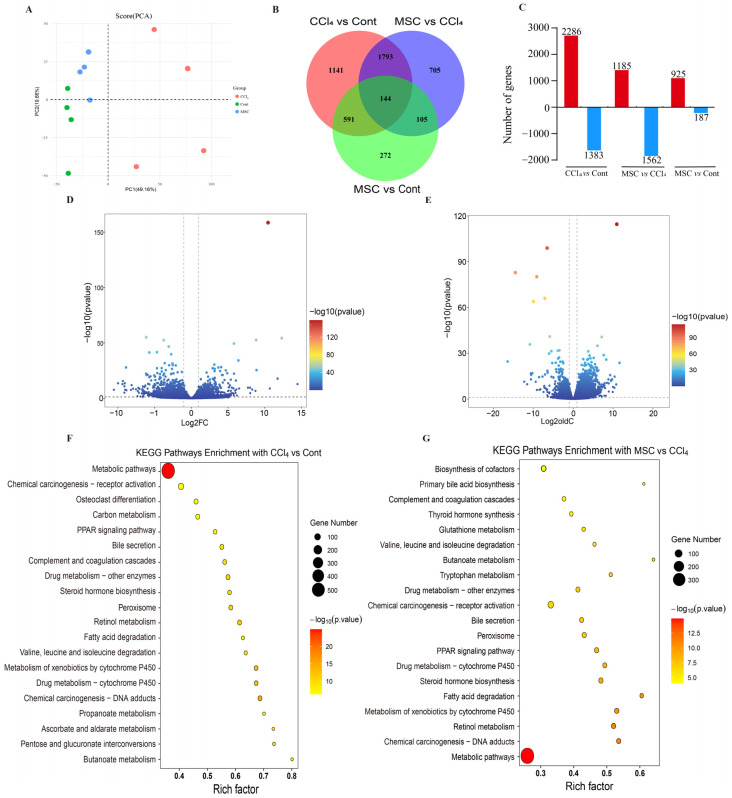
Changes in the liver transcriptional profile in CCl_4_-induced LF rats and ADMSCs transplantation rats. (**A**) PCA score plots of liver tissues of the three groups of rats; (**B**) VEEN plot of the number of DEGs in rat livers obtained via pairwise comparison between Cont, CCl_4_, and MSC groups; (**C**) Number of DEGs up- and downregulated in liver tissues revealed via pairwise comparisons; (**D**,**E**) Volcano plot of differential genes in CCl_4_ vs. Cont and MSC vs. CCl_4_; (**F**) Significantly enriched KEGG pathways in the CCl_4_ vs. Cont comparison; (**G**) Significantly enriched KEGG pathways in the MSC vs. CCl_4_ comparison. Cont, control group; CCl_4_, CCl_4_-treated group; MSC, ADMSC_S_-treated group; DEGs, differently expressed genes.

**Figure 4 ijms-24-16086-f004:**
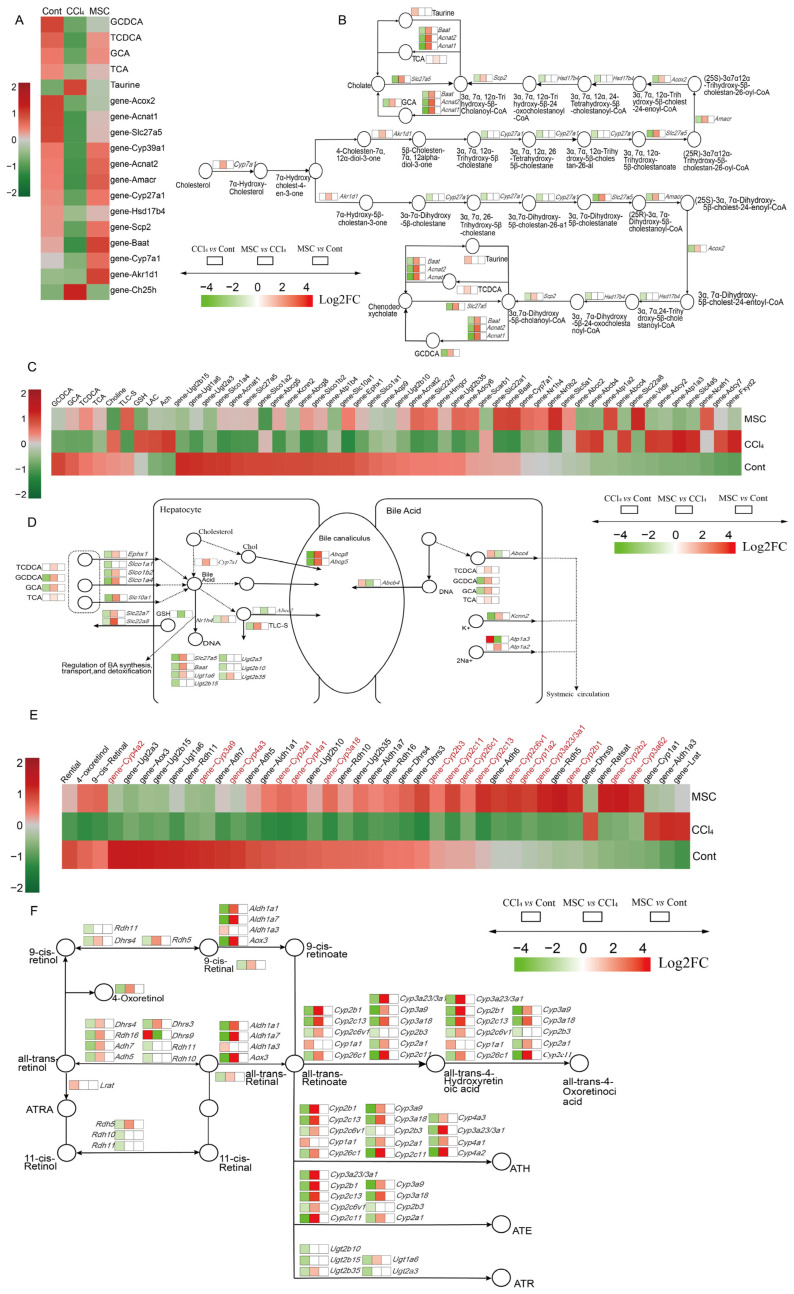
Effects of ADMSCs on genes and metabolites involved in primary BA synthesis, bile secretion, and retinol metabolism pathways. (**A**,**C**,**E**) Heatmaps of DMs and DEGs expression associated with primary BA biosynthesis, bile secretion, and retinol metabolism; (**B**,**D**,**F**) The most relevant metabolites and differentially expressed genes perturbed by CCl_4_ and reversed by ADMSCs in primary BA biosynthesis, bile secretion, and retinol metabolism pathway. DMs, differential metabolites; DGEs, differentially expression genes.

**Figure 5 ijms-24-16086-f005:**
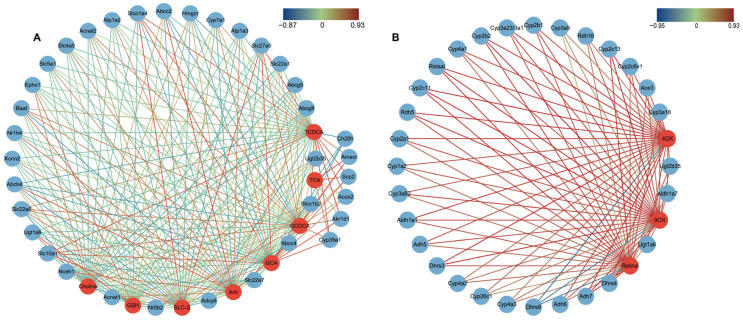
Correlation network analysis. (**A**) Correlation analysis of primary BA synthesis and bile secretion; (**B**) Correlation analysis of retinol metabolism. Red circles are DMs, and blue circles are DEGs. DMs, different expressed metabolites. DEGs, different expressed genes; TCDCA, Taurochenodeoxycholic acid; GCDCA, Glycochenodeoxycholic acid; GCA, glycocholic acid; TCA, Taurocholic acid; TLC-S, Taurolithocholic acid 3-sulfate; GSH, Glutathione; Ach, Acetylcholine; 9CR, 9-cis-Retinal; 4OX, 4-Oxoretinol.

**Figure 6 ijms-24-16086-f006:**
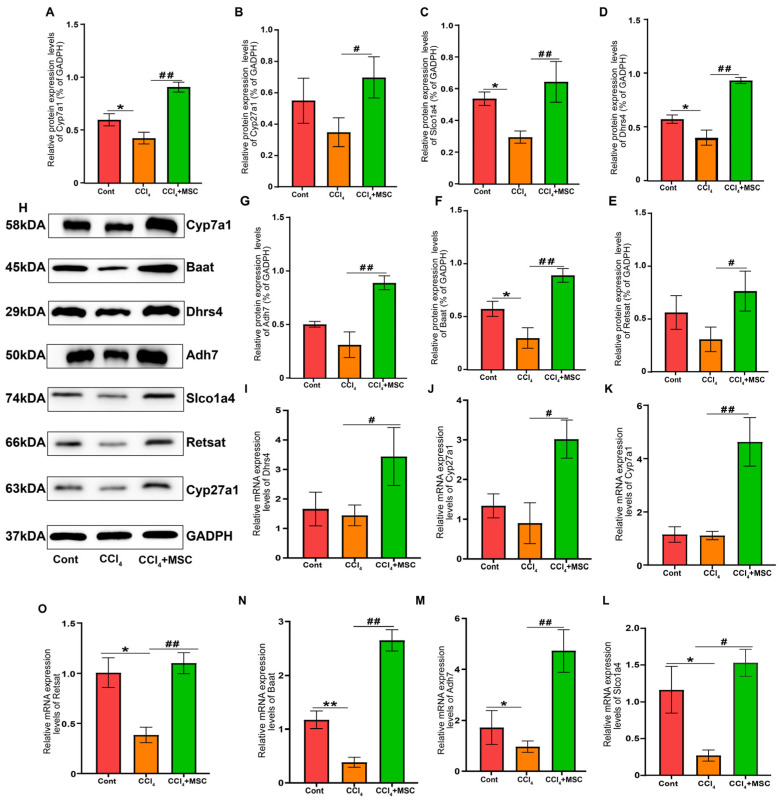
q-PCR and Western blot validation of 7 candidate genes. (**A**–**G**) The levels of *Cyp7a1*, *Cyp27a1*, *Baat*, *Dhrs4*, *Adh7*, *Slco1a4*, and *Retsat* mRNA expression in each group; (**H**–**O**) The levels of *Cyp7a1*, *Cyp27a1*, *Baat*, *Dhrs4*, *Adh7*, *Slco1a4*, and *Retsat* protein expression in each group. Each experiment was carried out at least three times, and data are displayed using mean ± SD. Cont, control group; CCl_4_, CCl_4_ intervention group; MSC, ADMSCs treatment group. *, *p* < 0.05; **, *p* < 0.01 vs. Cont group; ^#^, *p* < 0.05; ^##^, *p* < 0.01 vs. MSC group.

**Table 1 ijms-24-16086-t001:** Primers used for qPCR.

Gene	Primer Sequence (5-3′)	bp
*Cyp7a1*-Forward	CTGCCGGTACTAGACAGCAT	20
*Cyp7a1*-Reverse	TCCTCCTTAGCTGTGCGGAT	20
*Cyp27a1*	GGAACAGGTCAAGACCGACC	20
*Cyp27a1*	CTTGTTCAGCGCCTGGAG	18
*Adh7*	TGGGCCAGTTGATAACCCAC	20
*Adh7*	GGACAGTCCGAATGCTTTGC	20
*Baat*	CTGTCGAACTACGGTTTTGGC	21
*Baat*	GCTGTCAGCTTGGCCATTTT	20
*Retsat*	CATTCTGCCGAGCGTCTACT	20
*Retsat*	GCTGGGGGTTACTCCGTAAG	20
*Slco1a4*	AGCTTCTTCATAAAAACAGCAGTAA	25
*Slco1a4*	TGTTAATGCCAACAGAAACATCTTG	25
*Dhrs4*	CTTGGCACCTGGACTCATCA	20
*Dhrs4*	CTGGCTTGCCTAGCCTTCTA	20
*GADPH*	ACAGCAACAGGGTGGTGGAC	20
*GADPH*	TTTGAGGGTGCAGCGAACTT	20

## Data Availability

The data used to support the findings of this study are available from the corresponding author upon request.

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
