# Peer review of "Integrated Transcriptome and Metabolomics to Reveal the Mechanism of Adipose Mesenchymal Stem Cells in Treating Liver Fibrosis"

_ijms, 2023, doi:10.3390/ijms242216086_

Round 1

Reviewer 1 Report

Comments and Suggestions for Authors

Comments on the Quality of English Language

N/A

Reviewer 2 Report

Comments and Suggestions for Authors

Authors aimed to decipher underlying mechanism related to AMSC therapeutic effect in liver fibrosis using integrated omics on a preclinical model. However, there are numerous issued to consider before resubmit manuscript.

The choice of biological parameters to monitor fibrosis must be improved, AST, ALT are markers of hepatic cytolysis and not fibrosis. Indeed, in advance fibrosis plasma transaminase tends to decrease related to decrease of hepatocytes. There are other markers more specific to fibrosis such as Hyaluronic Acid, collagen …

Authors aimed (as mentioned in title) to perform “integrated” omics, here their only performed correlation between plasma metabolites and hepatic genes. Authors should use integrated omics tools such as Cytoscape to be able to identify new pathway of interest.

There are a specific focus on Bile Acids both in introduction and discussion, why mention it in introduction? Seems a bias in analysis, moreover other interesting pathway seems significant in statistical analysis (i.e amino acids). Moreover, a deeper literature review on BA and hepatic diseases in necessary. For example, no discussion about metabolic impact of BA in fibrosis, there is already known that BA could be involve in fibrosis such activation of their target receptor, FXR and TGR5. Hence, BA is not a novelty here and lead to lower interest of the manuscript.

Metabolomics method need to be described and detailed, what kind to mass spectrometry ? what library ? need to be detailed and clarify.

Table 1 and Table 2 seems to be useless in the main manuscript or not understandable

Some texts in figures are unreadable such as Figure 2, 4 ,5.

Comments on the Quality of English Language

RAS

Reviewer 3 Report

Comments and Suggestions for Authors

1) Part  should be more precise and description should be less popular science" Metabolomics is an emerging research method that involves analyzing endogenous 60m etabolites in urine, serum, and tissues to understand the metabolic pathways of an entire 61organ or system[18]. This approach provides valuable insights into the physiological or 62pathological processes caused by toxins or diseases[19]. Transcriptomics, on the other hand, 63focuses on predicting how external stimuli trigger responses in the body through changes 64in gene expression levels in tissues and organs[20]. 

2) why did you used this protocol for induction fibrosis ?"2.0ml/kg CCl4 diluted 4:6 (v/v) in olive oil twice a week for four 353weeks to induce liver fibros"  

3) why control group did not get medium (without cells) or olive oil ( without ccl4)? It is incorrect

4) you should add full name ALT, AST, and TBIL, IL-6 and TNF-α 

5) you should add in material and methods names of genes after rna seq in validation. You add information only in western blott

6) lack of information about isolation, characterization, phenotyping stem cells in material and methods and results 

7) It will be better to present results of genes expression as a table with arrows up and down, add information in which group there was a lower or higher expression

Round 2

Reviewer 2 Report

Comments and Suggestions for Authors

Unfortunately not original enough compared to existing literature.

Comments on the Quality of English Language

NA

Author Response

Dear Reviewer:

  We would like to express our deepest appreciation for your invaluable time and expertise dedicated to reviewing our manuscript titled [Integrated transcriptome and metabolomics to reveal the mechanism of adipose mesenchymal stem cells in treating liver fibrosis]. Your thorough and insightful feedback has immensely contributed to enhancing the quality and clarity of our work. After considerations of co-authors, we did our best for revising the manuscript according to your wisely suggestions.

Point1. We supplemented the molecular mechanisms of liver fibrosis in the introduction section, which can be seen in line 29-65 of the manuscript. At present, the reports of ADNSCs transplantation in the treatment of liver fibrosis mainly focus on controlling inflammatory response, promoting liver regeneration and immune regulation. However, whether it can be treated by correcting the disrupted metabolic pathways to exert its therapeutic effect is still unknown. Secondly, current studies only show that retinol metabolism disorder is accompanied by HSC activation, but its specific regulatory mechanism is still unclear. At the same time, the effect of BA on liver fibrosis is mainly focused on the change of serum BA and FXR and TGR5 signal transduction. There are few studies on liver bile acids. This study aims to determine whether ADMSCs can treat liver fibrosis by regulating metabolic pathways through the combination of transcriptome and metabolomics, and to find some new biomarkers for the treatment of liver fibrosis in the future.

Line 29-65: Any cause of chronic liver disease (hepatitis B and C, alcohol, cholestasis, and non-alcoholic steatohep-atitis (NASH)) leads to liver fibrosis[1]. Liver fibrosis is characterized by excessive synthe-sis and deposition of extracellular matrix (ECM)[2]. Hepatic stellate cells (HSC) are consid-ered major effectors of liver fibrosis because they are involved in ECM deposition and trigger immune responses by secreting cytokines and chemokines. In the process of HSC activation, on the one hand, HSC can stimulate the secretion of IL-17 in TH17 cells, up-regulate the expression of NF-κB and STAT3, and aggravate liver inflammation[3]. On the other hand, inhibition of the CXCR7-Id1 pathway through the fibroblast growth factor receptor 1 (FGFR 1) -CXCR4 axis inhibits normal liver regeneration and aggravate liver fi-brosis[4]. HSCs activation is accompanied by a large loss of retinoids, which further ag-gravates inflammation, fibrosis progression, and accelerates the occurrence of liver tumors. Unfortunately, the biological function of retinoid loss in HSCs and liver fibrosis remains unclear. Meanwhile, in liver injury, Cyp8a1 downregulation inhibits intrahepatic synthe-sis of bile acids and exacerbates liver fibrosis[5]. Inactivation of arnesoid X receptor (FXR) after liver injury leads to increased bile acid synthesis rate and fibroblast growth factor 19 (FGF19) expression in the liver, induces activation of inflammatory pathways and promotes hepatocyte apoptosis, leading to liver fibrosis[6, 7]. Therefore, inhibiting HSC activa-tion, regulating the level of body and immune response, and correcting metabolic disor-ders are effective ways to prevent or reduce liver fibrosis. Unfortunately, there is no generally accepted therapy for the treatment of liver fibrosis.

[1] HERNANDEZ-GEA V, FRIEDMAN S L. Pathogenesis of liver fibrosis [J]. Annual review of pathology, 2011, 6: 425-56.

[2]   SEKI E, BRENNER D A. Recent advancement of molecular mechanisms of liver fibrosis [J]. Journal of hepato-biliary-pancreatic sciences, 2015, 22(7): 512-8.

[3]   MENG F, WANG K, AOYAMA T, et al. Interleukin-17 signaling in inflammatory, Kupffer cells, and hepatic stellate cells exacerbates liver fibrosis in mice [J]. Gastroenterology, 2012, 143(3): 765-76.e3.

[4]   DING B S, CAO Z, LIS R, et al. Divergent angiocrine signals from vascular niche balance liver regeneration and fibrosis [J]. Nature, 2014, 505(7481): 97-102.

[5]   SONG Y N, ZHANG G B, LU Y Y, et al. Huangqi decoction alleviates dimethylnitrosamine-induced liver fibrosis: An analysis of bile acids metabolic mechanism [J]. Journal of ethnopharmacology, 2016, 189: 148-56.

[6]   KIR S, KLIEWER S A, MANGELSDORF D J. Roles of FGF19 in liver metabolism [J]. Cold Spring Harbor symposia on quantitative biology, 2011, 76: 139-44.

[7]        ZOLLNER G, MARSCHALL H U, WAGNER M, et al. Role of nuclear receptors in the adaptive response to bile acids and cholestasis: pathogenetic and therapeutic considerations [J]. Molecular pharmaceutics, 2006, 3(3): 231-51.

Point 2. We are very glad to receive your comments, and we have carefully modified the grammar, vocabulary and English of the full text. Please refer to the manuscript for details.

Point 3. For metabolomic analyses, we added references for detailed descriptions of the analyses.

Reviewer 3 Report

Comments and Suggestions for Authors

 Accept in present form. 

Author Response

Thank you for your guidance and assistance. We have made revisions to the article based on your feedback, which has a great guiding effect on us.